# Numerical Investigation of Mineral Grain Shape Effects on Strength and Fracture Behaviors of Rock Material

**Zhenhua Han [1,2,3,*], Luqing Zhang [1,2] and Jian Zhou [1,2]**

[1]  Key Laboratory of Shale Gas and Geoengineering, Institute of Geology and Geophysics, Chinese Academy of Science, Beijing 100029, China

[2]  Institutions of Earth Science, Chinese Academy of Sciences, Beijing 100029, China

[3]  College of Earth Science, University of Chinese Academy of Sciences, Beijing 100049, China

[*]  Correspondence: hanzhenhua@mail.iggcas.ac.cn; Tel.: +8610-8299-8644; Fax: +86-10-6201-0846

**Abstract:** Rock is an aggregate of mineral grains, and the grain shape has an obvious influence on rock mechanical behaviors. Current research on grain shape mostly focuses on loose granular materials and lacks standardized quantitative methods. Based on the CLUMP method in the two-dimensional particle flow code (PFC[2D]), three different grain groups were generated: strip, triangle, and square. Flatness and roughness were adopted to describe the overall contour and the surface morphology of the mineral grains, respectively. Simulated results showed that the grain shape significantly affected rock porosity and further influenced the peak strength and elastic modulus. The peak strength and elastic modulus of the model with strip-shaped grains were the highest, followed by the models with triangular and square grains. The effects of flatness and roughness on rock peak strength were the opposite, and the peak strength had a significant, positive correlation with cohesion. Tensile cracking was dominant among the generated microcracks, and the percentage of tensile cracking was maximal in the model with square grains. At the postpeak stage, the interlocking between grains was enhanced along with the increased surface roughness, which led to a slower stress drop.

**Keywords:** mineral grain shape; particle flow code; uniaxial compression simulation; rock mechanical property

## 1. Introduction

Rock is a natural heterogeneous material composed of a variety of minerals of different geometries, strengths, and deformation characteristics. Rock heterogeneity can be defined as the uneven changes of the mineral composition and microstructure in the spatial distribution influenced by the diagenesis and tectonics [1]. The macroscopic failure of rock is the gradual evolution of internal microcracks [2–4], so microheterogeneity significantly affects the macroscopic mechanical properties of rock [5–8]. Rock microheterogeneity mainly includes microstructure heterogeneity due to different grain shapes, sizes, and arrangements; elastic heterogeneity; and microcontact heterogeneity [9]. Heterogeneity is one of the fundamental reasons for rock strength differences and has always been an important research topic [10,11]. Affected by a diagenetic environment, the mineral grain shapes are often diverse and irregular, which has a great impact on rock mechanical properties [7]. Previous studies have explored the influences of mineral grain shape on rock macroscopic properties from a microscale perspective based on experiments and simulations.

Due to the complexity of the mineral grain shape, it is difficult to analyze the influence of mineral grain shape through rock prototype experiments. Therefore, self-made samples have often been used to study the grain shape effect. Shinohara et al. [12] found that the intergranular interlocking effect and

the internal friction angle of stainless-steel powder were increased with the increase of the particle edge angle by a triaxial compression test. Härtl et al. [13] studied the effect of grain shape on the shear behavior of a model using glass beads and found that the interlocking between different grain shapes could significantly increase the internal friction angle of the model. Liu et al. [14] quantified the grain shape of four different sand particles and glass spheres with the help of ImageJ software, which can easily achieve binary conversion of images. The results showed that the grain shape parameters (integral contour coefficient, sphericity, and angular angle) were well correlated with the friction angle and the dilatancy angle of the sand material. Johanson et al. [15] made samples from plastic pellets with different shapes (circular, heart shaped, and star shaped) and studied the effect of grain shape on the unconfined yield strength of the specimens by a shear test. The results showed that the number and direction of the contacts were key factors influencing the strength of the sample.

Because the information obtained from experiments is limited and it is difficult to make a sample composed of a complex grain shape, numerical simulations have often been used to study the effects of grain shape, as they can easily realize the formation of complex shapes by the combination of elements. Due to the heterogeneity, discontinuity, and anisotropy of rock materials, the theory of continuum mechanics has great limitations in simulating rock damage. As an important numerical simulation method, the discrete element method can effectively solve the mechanical problems of discontinuous material. Hence, the particle flow code (PFC) has also become an important tool for studying rock materials [16–20]. The traditional granule model in PFC is constructed from circular or spherical particles, but it can construct a cluster of particles of any shape using the CLUMP method [21]. In this method, two or more circular particles are joined together to form a complex shape, which can be regarded as a mineral grain. Based on this method, Santamarina and Cho [22] studied the effects of grain shape on the inherent anisotropy and the stress-induced anisotropy of sand materials. Shi et al. [23,24] studied the shear mechanical properties of nonrounded granular sands and found that the grain shape affected the peak strength, deformation characteristics, and the shear zone thickness of the specimen. Kong et al. [25] defined the shape coefficient using roundness and concavity to describe sand-like grains. Using biaxial and direct shear tests, negative linear correlations were found between the shape coefficient and the peak strength, internal friction angle, and shear strength of the sand material. Kerimov [26] investigated the effects of irregularly shaped grains on the porosity, permeability, and elastic bulk modulus of granular porous media, and they found that the grain shape had the greatest effect on permeability.

The above studies have revealed the relationship between the grain shape and the mechanical properties of loose granular material such as sand and soil. However, few papers have discussed the influence of mineral grain shape on rock mechanical behavior. Unlike loose material, the mineral grains of rock are bonded together, hence the grain shape effects are different with loose material. In the study of rock mineral shape effects, quantitative descriptions of grain shape characteristics are essential. Because of the irregularity of the mineral grain shape, studies on the relationship between the shape parameters and rock macroscopic mechanical properties are lacking. Based on PFC, Cho and Martin [21] concluded that the models with complex grain shapes have higher tensile strength to compressive strength ratio compared with the circular grain model. Kem et al. [27] thought that the oblate plate grain shape and corresponding shape orientation distributions would lead to an increase in the calculated elastic anisotropy of rocks. Rong et al. [7] investigated the effects of grain shape on rock mechanical properties using three-dimensional PFC ($PFC^{3D}$). He used the sphericity index to quantitatively characterize the grain shape and found that the initiation, damage, and peak strength all decreased with an increasing sphericity index. However, it is not comprehensive to quantify the effect of grain shape on rock mechanical properties only using the indicator of sphericity because sphericity only reflects the proximity of the grain to the sphere and it cannot reflect the roughness of the grain surface. Thus, there is a need for further investigation of the mineral grain shape effects on rock mechanical properties.

The specific problem we addressed in the present work is a preliminary investigation of the irregular grain shape effects on the rock strength and fracture behaviors. Lac du Bonnet (LDB)

granite was used for model calibration, which is a candidate host rock for the repository of high-level radioactive waste. Previous studies on grain shape are limited to a specific set of four or five irregularly shaped grains. In the present work, three different grain groups were constructed based on the CLUMP method in two-dimensional PFC (PFC$^{2D}$), respectively strip, triangle, and square. In each shape group, a series of six grains was included for systematic investigation. Combined with the grain characteristics, the shape parameters, which can describe the overall contour and surface morphology of the mineral grain, were defined. The influences of mineral grain shape on rock mechanical properties under uniaxial compressive conditions were further analyzed.

## 2. Modeling Methodology

PFC$^{2D}$ is based on the distinct element theory, which was first proposed by Cundall [16]. It is suitable for analyzing material mechanics problems under quasi-static and dynamic conditions. In this method [17], the material is represented as an aggregate of numerous rigid circular particles. Noncontinuous mechanical behaviors of rock material can be observed as a result of the interactions and movements of the rigid particles. The interaction of the particles is treated as a dynamic process with states of equilibrium developing whenever the internal forces balance. The calculation process can be seen from Figure 1. Newton's second law was used to calculate the translational and rotational motion of each particle arising from the forces acting upon it, and the force–displacement law was used to update the contact forces arising from the relative motion at each contact. A time-step algorithm was used repeatedly on each particle. For further details, please refer to Potyondy and Cundall [17].

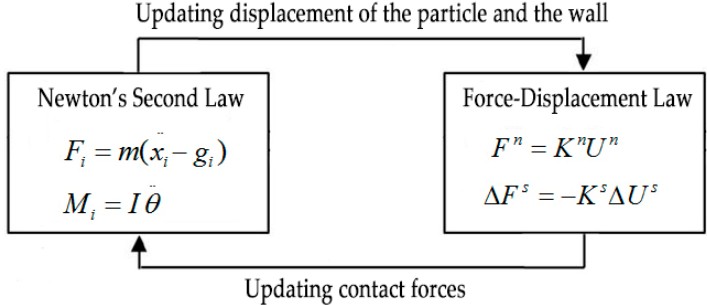

**Figure 1.** Calculation principles of the particle discrete element method [1].

PFC provides three basic contact models: parallel bond model, contact bond model, and sliding model. The parallel bond model is among the most frequently used models in rock mechanics research. The parallel bond model can be assumed to be a cemented polymer (Figure 2), where particles are cemented together at the point of contact so as to provide certain resistance to the exterior load. The contact force between particles is represented by $F_i$. The force and moment carried by the parallel bond are represented by $\overline{F}_i$ and $\overline{M}_i$, denoted in Figure 2. When an external force acting on each parallel bond exceeds its limit shear or tensile strength, the bond is damaged, and a shear or tensile microcrack develops at the corresponding position. With the ongoing generation of microcracks, a macrofracture can be formed by the linking of these individual microcracks.

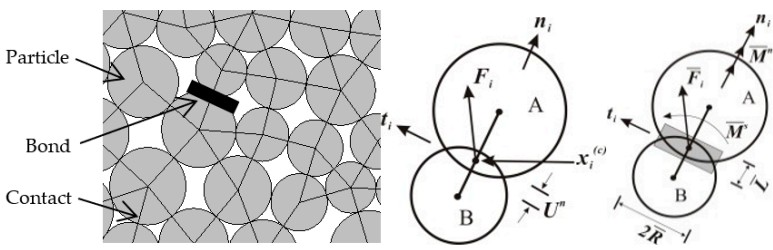

**Figure 2.** Force–displacement behavior of the grain-bonding system after [17].

## 3. Model Descriptions

### 3.1. Construction of Mineral Grains of Different Shapes

The basic particle shape is a circle, but noncircular grains can be achieved by the combination of two or more circular particles using the CLUMP method. In this method, grains with irregular shapes can be represented as single objects, which means the particles within a clump may overlap to any extent, but contact forces are not generated between these particles. The mineral grain is internally equivalent to a rigid body, and there is only a slight deformation at the contact boundary between the mineral grains. Considering the complexity of real mineral grains, it is difficult to simulate each mineral grain shape. Therefore, some simple grain shapes were adopted in this study. As shown in Figure 3, three mineral grain groups of different shapes were constructed: strip, triangle, and square. There were six grains of similar shapes in each group.

1. The strip-shaped grain was composed of two circular particles of the same radius $R$. The six grains in this group were represented as S1–S6. From S1 to S6, the distance $d$ between the centers of the two circular particles gradually increased, and the ratios of $d$ to $R$ were 0, 0.25, 0.5, 1.0, 1.25, and 1.5, respectively.
2. The triangular grain was composed of three circular particles of the same radius $R$. The line connecting the centers of the three circular particles was an equilateral triangle with a side length of $d$. The six grains in this group were represented as T1–T6. From T1 to T6, the value of $d$ was gradually increased. The variation of $d/R$ was consistent with the strip-shaped grains.
3. The square grain was composed of four circular particles of the same radius $R$. The line connecting the centers of the four circular particles was a square with a side length of $d$. The six grains in this group were represented as R1–R6. From R1 to R6, the value of $d$ was gradually increased. The variation of $d/R$ was also consistent with the strip-shaped grains.

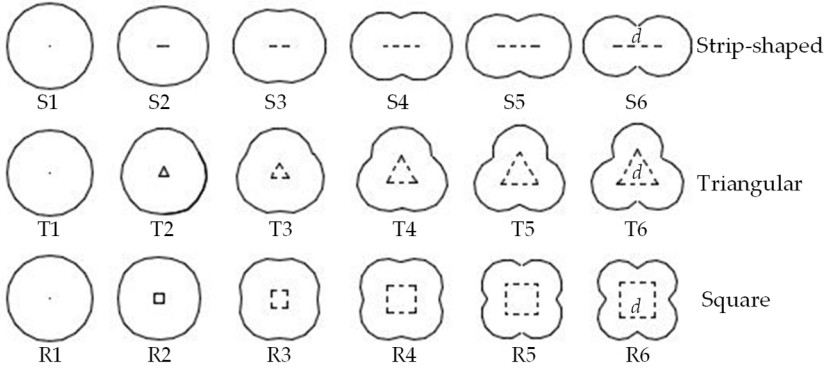

**Figure 3.** The construction of three particle shapes using the CLUMP method.

Taking the strip-shaped grain as an example, the circular particle S1 could be directly generated, but grains S2–S6 needed to be generated by the CLUMP method of PFC$^{2D}$. In the CLUMP method, the circular particles were first created, then the circular particles were replaced with grains S2–S6. The replacements followed three principles: "area equivalence", "area center equivalent", and "directional randomization". This means that the area and centroid of grains S2–S6 were the same as grain S1, but their orientations were random.

### 3.2. Quantitative Description of Grain Shape

For an irregular grain, the shape can be described from two aspects [14]. First is the overall contour, which can be simply described as circular, square, plate, or column. The second aspect is the surface morphology, which refines the grain outline based on the first aspect. It focuses on the

irregularity of the grain surface or the fluctuation of the grain boundary. Flatness was adopted in this study to quantify the overall contour of the grain, and the surface morphology of the grain was described by roughness [28,29]. As shown in Equation (1), flatness is the ratio of the maximum Feret's diameter of the grain to the minimum Feret's diameter. Feret's diameter means the vertical distance between two parallel lines along the grain boundaries, as shown in Figure 4. Flatness characterizes the elongation property of the grain with a minimum value of 1. The grain approaches are spherical or circular when this value is closer to 1. On the contrary, the larger the flatness value, the flatter and narrower the grain. As shown in Equation (2), roughness is defined as the square of the ratio of two perimeters, which are the perimeter of the grain itself and the perimeter of the smallest circumscribing polygon of the grain (Figure 4). Roughness characterizes the fluctuation of the grain boundary curve, and greater roughness means a more irregular surface morphology of the grain.

$$e = L/B; \tag{1}$$

$$r = (P/PC)^2; \tag{2}$$

where $e$ is the flatness, and $L$ and $B$ are the maximum and minimum Feret's diameters, respectively. Roughness is expressed by $r$, $P$ is the perimeter of the grain, and $Pc$ is the perimeter of the smallest circumscribing polygon of the grain.

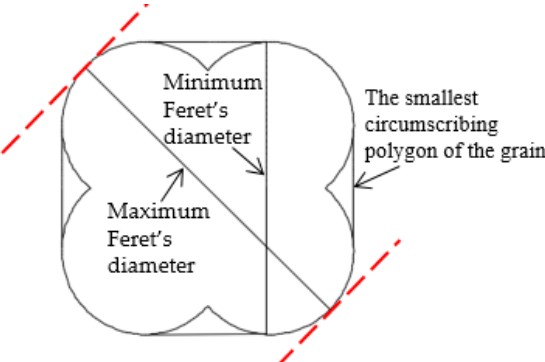

**Figure 4.** Basic dimension parameters of irregular grains used for calculating flatness and roughness.

According to Equations (1) and (2), the flatness and roughness of the three mineral grain groups mentioned above were calculated, and the results are shown in Figure 5. As can be seen from Figure 5a, the flatness of the strip-shaped grain group was significantly higher than that of the triangular and square grain groups. In the strip-shaped grain group, from S1 to S6, the flatness increased rapidly with the increase of $d/R$. The flatness of the square grain group was slightly higher than that of the triangular grain group, and the flatness increased slightly from T1 to T6 and R1 to R6. It can be seen from Figure 5b that the square grain group had the highest surface roughness, followed sequentially by the triangular and the strip-shaped grain group. With the increase of $d/R$, the roughness of the three grain groups all increased. When the $d/R$ value was low (shape numbers 1–3), an increase in the roughness was not obvious; however, it increased faster when the $d/R$ value was higher (shape numbers 4–6).

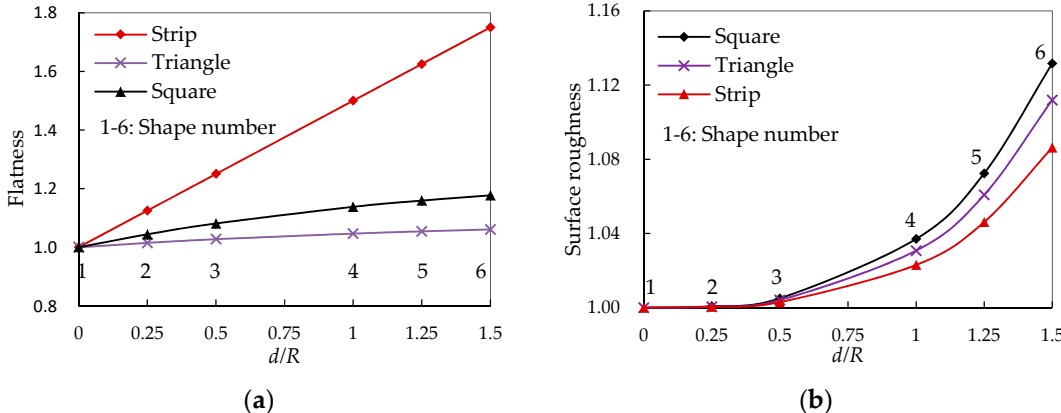

**Figure 5.** Shape factor variation of three representative grain groups: (**a**) flatness; (**b**) roughness.

### 3.3. Calibration of the Microparameters

The microparameters used in PFC$^{2D}$ cannot be directly acquired from experimental laboratory tests, which need to be calibrated using a trial-and-error procedure. In this procedure, the microparameters needed to be optimized continuously until the simulated macro parameters matched the experimental results. The macro parameters used for calibration generally include the elastic modulus, Poisson's ratio, uniaxial compressive strength (UCS), and tensile strength. As shown in Figure 6, the circular-shaped grain was used for microparameter calibration, and the model was 50 mm in width and 100 mm in height, with the particle radii ranging from 0.5 to 0.83 mm following a normal distribution. The walls were used to apply a stress-boundary condition, and the wall velocity was controlled by a numerical servo-control mechanism in order to maintain a specified wall stress.

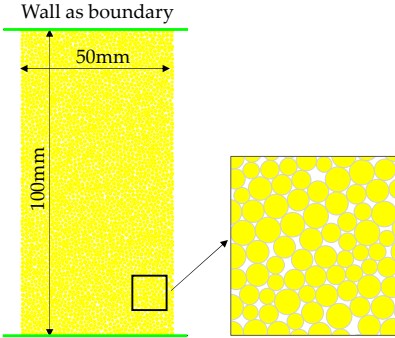

**Figure 6.** Numerical model for the uniaxial compression simulation.

In this study, the microparameters were calibrated to match the macroproperties of the uniaxial compression test of Lac du Bonnet (LDB) granite, including the elastic modulus, peak strength, and Poisson's ratio, which were 70 GPa, 224 MPa, and 0.26, respectively [9]. LDB granite is a representative brittle rock. Because of its comprehensive rock mechanics parameters, it is often used for model validation [9,18,30]. After a trial-and-error process, the elastic modulus, peak strength, and Poisson's ratio of the calibration model were determined to be 70 GPa, 207 MPa, and 0.26, respectively, which were consistent with the corresponding experimental results. The mesoscopic physical and mechanical parameters of the model are shown in Table 1. As can be seen, it includes the micromechanical parameters of the particles and the parallel bond between them. According to the existing literature [31,32] and our calibration process, the macro elastic modulus of the model was controlled by the contact modulus, and Poisson's ratio was affected by the ratio of normal stiffness to shear stiffness. The strength of the model was mainly decided by the tensile and shear strengths of the bond.

**Table 1.** Micro-physico-mechanical parameters of the particle and parallel bond.

| Classification | Microparameters | Notations | Values |
|---|---|---|---|
| Particle | Density (kg/m$^3$) | | 2630 |
| | Contact modulus (GPa) | $E$ | 62 |
| | Normal-to-shear stiffness ratio | $k_n/k_s$ | 2.5 |
| | Friction coefficient | $u$ | 0.5 |
| Parallel bond between particles | Tensile strength (MPa) | $\sigma_c$ | 157 |
| | Standard deviation /MPa | $\sigma_{cs}$ | 36 |
| | Normal-to-shear stiffness ratio | $\overline{k_n}/\overline{k_s}$ | 2.5 |
| | Shear strength (MPa) | $\tau_c$ | 175 |
| | Standard deviation /MPa | $\tau_{cs}$ | 40 |
| | Modulus (GPa) | $E_c$ | 62 |
| | Radius multiplier | $\lambda$ | 1 |

## 4. Analysis of Grain Shape Effects

A series of numerical models were established based on the grain shapes shown in Figure 3. The microparameters shown in Table 1 were adopted to conduct uniaxial compression simulations. According to the results, the influences of grain shape on the macroscopic parameters and failure mode of rock were analyzed.

### 4.1. Effects of Grain Shape on Rock Macroscopic Parameters

Figure 7 shows the effects of grain shape on rock peak strength and elastic modulus. It can be seen that the peak strength and elastic modulus of the model with the strip-shaped grain was the largest, followed by the triangular grain and square grain models. Compared with the circular grain (shape number 1), the elastic modulus of the models with clumped grains was significantly increased (shape numbers 2–6). According to our simulated results, the effects of grain shape on rock peak strength and elastic modulus were closely related to the model's porosity. Figure 8 shows the effects of grain shape on porosity. It can be seen that the porosity of the square grain model was the largest, followed by the triangular grain and strip-shaped grain models, contrary to the variation of peak strength and elastic modulus.

Numerous experiments have shown that increased porosity can reduce rock strength and elastic modulus [33–35]. Hudyma et al. [36] studied the mechanical properties of porous tuff and found that the compressive strength and elastic modulus of tuff decreased with increasing porosity. Al-Harthi et al. [37] studied the effect of pores on the mechanical properties of basalt based on image analysis techniques and came to a similar conclusion as Hudyma et al. Using PFC$^{3D}$ simulations, Schopfer et al. [38] found that rock strength and elastic modulus decreased with increasing porosity. Therefore, the macroscopic mechanical parameters decreased with the increase of the porosity in this study. It should be noted that the porosity of a model in PFC is a set value, which was set to 0.16 in our simulation. However, in the process of adjusting the model's internal stress, the porosity of the model will change until the model reaches an isotropic stress state, especially when the grain shape is complicated. It can be seen from Figure 8 that the actual porosity of the model was approximately 0.16 when the grain shape was circular (shape number 1), which was in accordance with the set value. However, when clumped grains were used, the actual porosity of the model varied greatly from the set value, which indicated that the grain shape affected the model porosity and further affected rock elastic modulus and peak strength.

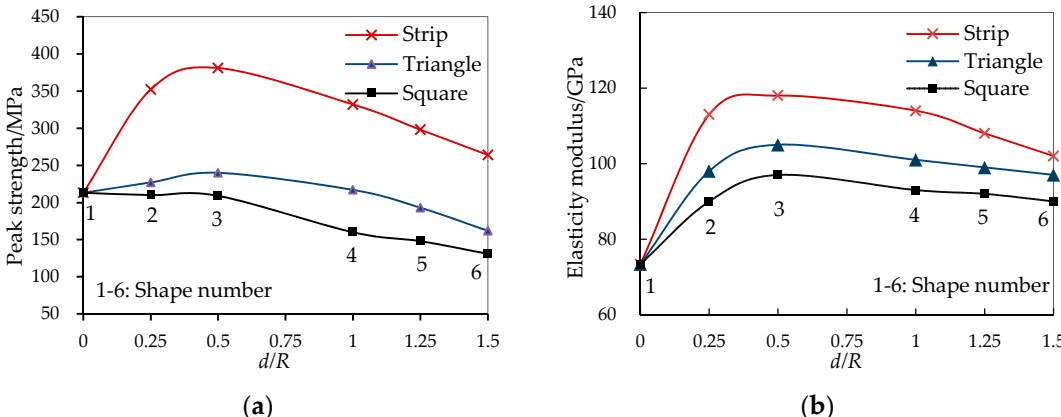

**Figure 7.** Effects of grain shape on rock (**a**) peak strength and (**b**) elastic modulus.

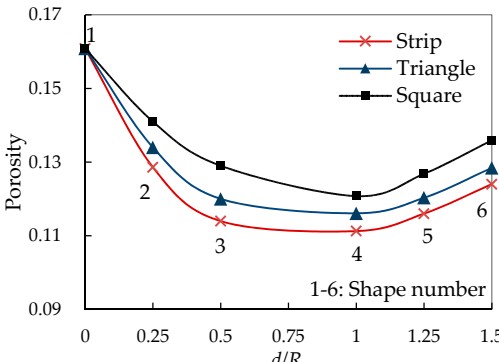

**Figure 8.** Effects of grain shape on rock porosity.

It can also be seen from Figure 7a that with the increase of the $d/R$ value, the peak strength of the models with strip-shaped and triangular grains first increased and then decreased, but it only decreased when the grain shape was square. This variation was closely related to the flatness and roughness of the mineral grain. Firstly, as can be seen in Figure 5a, the flatness of the strip-shaped grain was obviously higher than that of the triangular and square grains. Also, the peak strength of model with strip-shaped grains was significantly higher, indicating that an increase in flatness can enhance rock peak strength, which was mainly because the strip-shaped grains were difficult to rotate and displace when they were assembled together. Zhang et al. [39] also thought that a model with strip-shaped grains would have higher strength. Secondly, because the flatness of the square and triangular grains was significantly lower (Figure 5a), the overall contour was close to circular. Therefore, the peak strength was slightly affected by flatness in the models with square and triangular grains. However, the grain surface roughness was relatively higher, and the main influencing factor affecting the peak strength was roughness. As can be seen from Figure 7a, the peak strength decreased with increasing roughness. Therefore, the variations of rock peak strength shown in Figure 7a were due to the combined effects of mineral flatness and roughness. The effects of shape factors on the peak strength of models with different grain shapes are discussed in detail as follows:

1.  Strip-shaped grain

It can be seen from Figure 5 that when the $d/R$ value was low, the surface roughness of the strip-shaped grain increased slowly, but the flatness increased rapidly. Therefore, the obvious increase of peak strength from models S1 to S3 was mainly controlled by the flatness. When the $d/R$ value became higher, the roughness increased rapidly, and the peak strength decreased significantly, indicating that the peak strength was mainly affected by the roughness from models S4 to S6. It can be concluded that when the roughness exceeds a certain limit, it will be the main factor of strength change.

2. Triangular grain

When the *d/R* value was low (shape numbers T1–T3), similar to the model with the strip-shaped grain, the increase of peak strength of models with triangular grains was also controlled by the flatness. However, because the increase of the flatness of triangular grain was much lower than that of the strip-shaped grain, the increase of peak strength of models with triangular grains was also obviously lower than that of the models with strip-shaped grains. When the *d/R* value was higher (shape numbers T4–T6), the decrease of the peak strength was affected by the increasing roughness.

3. Square grain

For the models with square grains, the decrease of the peak strength was controlled by the roughness. The roughness increased slowly from models R1 to R3, which led to the slow decrease of the peak strength. The increase of the roughness was accelerated from models R4 to R6; thus, the decrease of the peak strength was also significantly reduced.

In addition, it was found that the grain shape also had effects on the cohesion and internal friction angle of the model. These are two important parameters for characterizing the mechanical properties of mineral grains. Generally, the greater grain surface roughness means a higher internal friction angle. In bulk materials such as sand, samples with complex grain shapes generally have higher strength, which is mainly because the interlocking between mineral grains can increase the internal friction angle [12,13,40]. In other words, the strength of the bulk material is positively correlated with the internal friction angle. However, in rock materials, the strength of rock was found to be positively correlated with cohesion. Taking the model with circular grains (shape number 1) as an example, as shown in Figure 9, the peak compressive strengths of the model under three confining pressures (0, 50, and 200 MPa) were obtained. Three ultimate stress circles and their common tangent line were further obtained according to the confining pressure and the corresponding peak compressive strength. The intersection of the common tangent line and the $\tau$ axis was the cohesion of the model. Figure 10 shows the effects of grain shape on rock cohesion. It can be seen that the effects of grain shape on rock cohesion were similar to the effects on rock uniaxial peak strength. In PFC models, the mineral grains are bonded together, and from a macroscopic view, the cohesion controls the strength of the model. Here, the peak strength of the rock increased with the increase of the cohesion.

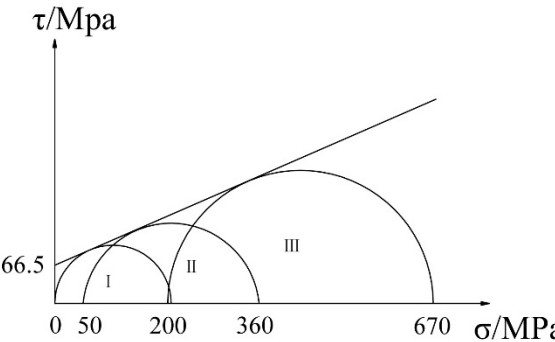

**Figure 9.** Cohesion of the model with circular grains (shape number 1).

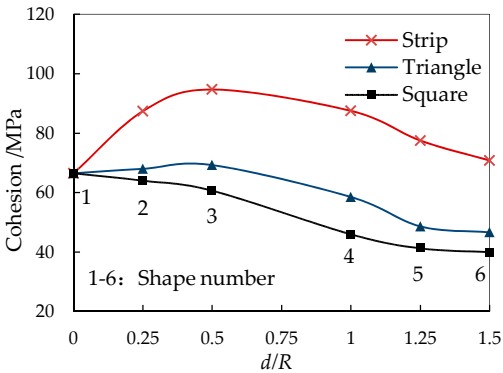

**Figure 10.** Effects of grain shape on rock cohesion.

### 4.2. Effects of Grain Shape on Rock Failure Mode

The macroscopic failure of rock is a gradual evolutionary process of internal microcracks, and the microcracking behavior has a great influence on the ultimate failure mode of rock. Figure 11 presents the spatial distributions of microcracks for numerical models with different grain shapes when the models reached the postpeak stage and the axial stress was 80% of the peak strength. The generated microcracks can be divided into two types—tensile and shear cracks—represented by black and red segments, respectively. A large number of previous studies [41–43] have shown that tensile failure plays a dominant role in rock failure mechanisms. In this study, the microcracks were dominated by tensile cracks (Table 2), accounting for more than 80% of the total number of cracks. It can also be seen from Table 2 that the percentage of tensile cracks was maximal in the model with square grains, followed by the triangular grain and strip-shaped grain models.

**Table 2.** Percentage of tensile cracks of rock with three particle shapes in postpeak stage $\sigma = 0.8\ \sigma_c$.

| Ratio of Tensile Crack (%) | Shape Number | | | | | |
|---|---|---|---|---|---|---|
| | 1 | 2 | 3 | 4 | 5 | 6 |
| Strip-shaped grain | 88 | 87 | 83 | 85 | 79 | 83 |
| Triangular grain | 88 | 92 | 90 | 92 | 92 | 90 |
| Square grain | 88 | 92 | 95 | 93 | 93 | 91 |

Figure 12 shows the simulated stress–strain curves of the models with triangular grains (taking the models with shape numbers T3, T4, and T6 as examples). It can be seen that the stress dropped after the peak strength gradually became slower from T3 to T6. This can be explained by the fact that from T3 to T6, the surface roughness of the grain increased, and the irregularity of the grain shape increased. In the pre-peak stage, the mineral grains were cemented together. The bond damage increased after the rock reached the peak strength, then the number of microcracks increased rapidly and formed macrocracks. The mineral grains on both sides of the crack began to move around each other along the grain surface and produced significant displacement. In mineral grains with a high surface roughness, the interlocking effect and friction between grains were stronger, which made samples have a higher residual strength and resulted in slower stress drop. Figure 13 shows the stress–strain curves of the models with three grain shapes (taking the models with shape numbers S6, T6, and R6 as examples). As can be seen, from the strip-shaped grain to the square grain, the surface roughness increased, and the postpeak stress drop of the three models also became slower, which further confirmed the influence of grain shape on rock characteristics after peak strength.

**Figure 11.** Failure mode of rock with three grain shapes in the postpeak stage when σ = 0.8 σc (black segments represent tensile cracks and red segments represent shear cracks).

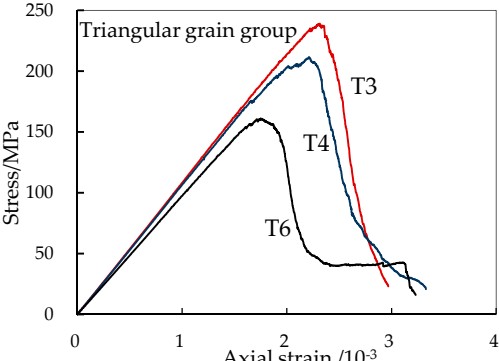

**Figure 12.** Stress–strain curves of different triangular grain models (case study of models T3, T4, and T6).

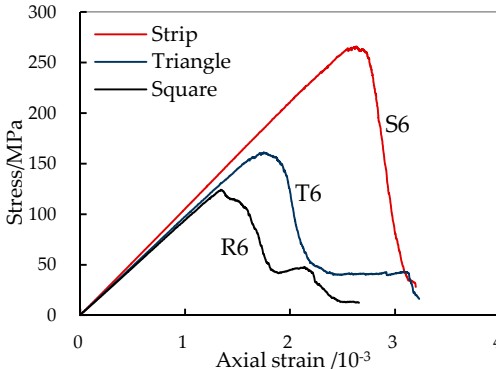

**Figure 13.** Stress–strain curves of three different grain shape models (case study of model with shape number 6).

According to the above analysis, the PFC model was composed of grains by cementation, and the failure of the model was essentially the break of the intergrain bond. Hence, the peak strength of the model was macroscopically controlled by cohesion. After the axial stress reached the peak strength of the model, obvious displacement started to appear along the failure surface. The interlocking between grains played a major role, the relative sliding between the grains became difficult with the increasing surface roughness, and the macroresponse was that the stress drop became slower.

## 5. Conclusions

In this study, the effects of mineral grain shape on the mechanical properties of LDB granite were investigated by means of uniaxial compression simulations. Based on the CLUMP method in PFC$^{2D}$, three different grain shapes were constructed: strip, triangle, and square. The relationship between the grain shapes and rock mechanical properties was analyzed, and the main conclusions are as follows:

1. The elastic modulus and peak strength of rock are affected by the mineral grain shape. In our study, the model with strip-shaped grains had the largest elastic modulus and peak strength, followed by the triangular grain and square grain models. The mechanism of grain shape effect on the peak strength and elastic modulus was mainly achieved by affecting the porosity of the model. The increasing porosity led to a smaller peak strength and elastic modulus.

2. Flatness and roughness can describe the overall contour and surface morphology of mineral grains well. The effects of grain flatness and surface roughness on rock peak strength are the opposite. A flat grain helps to increase rock strength, while rock strength is reduced due to increased porosity if the grain has higher roughness. Further, the relationship between the peak strength and the cohesion of rock is positively correlated.

3. Here, the generated microcracks were mainly of the tension type, and the model with square grains had the maximum ratio of tensile cracking, followed by the triangular grain and strip-shaped grain models. The shape of the mineral grain also had a significant effect on rock mechanical behaviors at the postpeak stage. The stress drop became slower with an increasing surface roughness because the interlocking restrained the slip and rotation of grains on the fracture surface.

However, there were some defects in the modeling. The grain shapes adopted in this study were simple and single, while real mineral grain shapes are very complex. Since the CLUMP method can realize the construction of complex grain shapes, future works should connect this study to real rocks. In addition, three-dimensional grain shape effects should also be investigated in future studies.

**Author Contributions:** Z.H. and L.Z. designed the theoretical framework; Z.H. wrote the basic code of the program and designed the numerical simulation; J.Z. checked the simulation results.

**Funding:** This research was funded by the National Natural Science Foundation of China (Grant Nos. 41572312, 41672321), China Postdoctoral Science Foundation (Grant Nos. 2018M630204, 2019T120133), and China Scholarship Council (Grant No. 201804910293) to the third author.

**Conflicts of Interest:** The authors declare no conflict of interest.

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
