# Peer review of "Numerical Investigation of Mineral Grain Shape Effects on Strength and Fracture Behaviors of Rock Material"

_applsci, doi:10.3390/app9142855_

Round 1

Reviewer 1 Report

The authors have studied the effects of mineral grain shape on the mechanical properties of Lac du Bonnet (LDB) granite by using the CLUMP method in the particle flow code (PFC) and uniaxial compression simulation.

The subject is relevant to the scope of the journal and the paper is very well organized. The work is original and there is a significant amount of new work in the paper. The paper is recommended for publication after addressing the following comments.

-       All the papers published on the influence of mineral grain shape on rock mechanical behavior should be addressed.

-       Please mention the reference for Figure 1.

-       Add a figure showing the model including particles, dimensions and boundary conditions.

Author Response

Dear Reviewer:

Thank you very much for the time and effort you expend on this paper. We found the comments and recommendations most helpful and have revised the manuscript. The main corrections in the paper and the responds to your comments are in the attachment.

All amendments have been highlighted in yellow in the modified article. We hope that our revised version will be satisfactory.

Thanks and Best regards.

Sincerely yours,

Zhenhua Han on behalf of the authors.

Reviewer 2 Report

The contribution and novelty of the research must state in clear way. What is the new methods and findings in the study? How you apply the results to the real samples? 

The methodology is not clear enough. Do you provide the result only based on generated models from PFC? Is there any verification?

The reference format in the text is confusing.

How you calibrate the model?

Author Response

(The authors gave the same response as above.)
